# A Narrative Review on Maternal Choline Intake and Liver Function of the Fetus and the Infant; Implications for Research, Policy, and Practice

**DOI:** 10.3390/nu16020260

**Published:** 2024-01-15

**Authors:** Rima Obeid, Christiane Schön, Emma Derbyshire, Xinyin Jiang, Tiffany J. Mellott, Jan Krzysztof Blusztajn, Steven H. Zeisel

**Affiliations:** 1Department of Clinical Chemistry and Laboratory Medicine, Saarland University Hospital, D-66420 Homburg, Germany; 2BioTeSys GmbH, Nutritional CRO, Schelztorstrasse 54-56, D-73728 Esslingen, Germany; 3Nutritional Insight, Surrey KT17 2AA, UK; 4Department of Health and Nutrition Sciences, Brooklyn College, City University of New York, 4110C Ingersoll Hall, 2900 Bedford Ave., Brooklyn, NY 11210, USA; 5Department of Pathology & Laboratory Medicine, Boston University Chobanian & Avedisian School of Medicine, Boston, MA 02118, USA; 6Department of Nutrition, University of North Carolina, Chapel Hill, NC 27514, USA

**Keywords:** dietary choline, liver function, fetus, infant, lactation, pregnancy

## Abstract

Dietary choline is needed to maintain normal health, including normal liver function in adults. Fatty liver induced by a choline-deficient diet has been consistently observed in human and animal studies. The effect of insufficient choline intake on hepatic fat accumulation is specific and reversible when choline is added to the diet. Choline requirements are higher in women during pregnancy and lactation than in young non-pregnant women. We reviewed the evidence on whether choline derived from the maternal diet is necessary for maintaining normal liver function in the fetus and breastfed infants. Studies have shown that choline from the maternal diet is actively transferred to the placenta, fetal liver, and human milk. This maternal-to-child gradient can cause depletion of maternal choline stores and increase the susceptibility of the mother to fatty liver. Removing choline from the diet of pregnant rats causes fatty liver both in the mother and the fetus. The severity of fatty liver in the offspring was found to correspond to the severity of fatty liver in the respective mothers and to the duration of feeding the choline-deficient diet to the mother. The contribution of maternal choline intake in normal liver function of the offspring can be explained by the role of phosphatidylcholine in lipid transport and as a component of cell membranes and the function of choline as a methyl donor that enables synthesis of phosphatidylcholine in the liver. Additional evidence is needed on the effect of choline intake during pregnancy and lactation on health outcomes in the fetus and infant. Most pregnant and lactating women are currently not achieving the adequate intake level of choline through the diet. Therefore, public health policies are needed to ensure sufficient choline intake through adding choline to maternal multivitamin supplements.

## 1. Introduction

The United States National Academy of Medicine (NAM) considered choline an essential nutrient for adults because the endogenous synthesis of choline in the liver is not sufficient to maintain normal body functions [1]. A choline-deficient diet providing only 13 mg of choline per day can cause elevated serum alanine aminotransferase activity in healthy adult men within 3 weeks, suggesting incipient liver damage [2]. This effect was averted when the participants received 500 mg/d of choline after the depletion phase [2]. Several studies since 1990 provided evidence that dietary choline is needed for normal liver function of adults (reviewed in [1,3]).

The dietary reference value of choline for non-pregnant women is 425 mg/d according to NAM [1] and 400 mg/d according to the European Food Safety Authority (EFSA) [3]. The reference values for pregnant and lactating women are 450 mg/d and 550 mg/d, respectively, according to NAM [1] and 480 and 520 mg/d according to the EFSA [3]. An additional intake of 50 mg/d to 120 mg/d during pregnancy and lactation is assumed to ensure choline supply to the fetus or the infant. Judgement of the intake levels during pregnancy and lactation was theoretical and was not linked to any health outcome in the child. Evidence from studies in adults [2] may suggest that maternal choline intake could have a role in liver function of the fetus and infant. To the best of our knowledge, no studies have been published on this topic since 1990.

In the present review, we searched for evidence on whether insufficient dietary choline intake by the mother may cause fatty liver in the fetus and infant. Implications of this knowledge on future research are discussed. In addition, adequacy of dietary choline intakes during pregnancy and lactation compared to the present recommendations and possible mitigation strategies are discussed. A non-systematic search in PubMed and Google Scholar (between 1930 and 1990) was conducted using the following key words: “maternal choline and foetal development”, “maternal choline and fetal liver”, “maternal choline deficiency and fatty liver”, “maternal choline intake and fatty liver”, “choline deficiency in pregnancy and fatty liver”, “choline deficiency during lactation and fatty liver”, “choline and foetal liver”, “choline metabolism in the fetus”, and “choline deficiency during pregnancy and newborn health”. The reference lists of the relevant articles were also screened.

## 2. Choline Sources in the Diet

Fish, meat, egg, and dairy products, in addition to some plant-source foods, such as soy beans, are rich in choline and choline derivatives. The relative contribution of an animal diet to the daily intake of choline is generally larger than that of a plant-based diet. Different vegetarian diets (e.g., vegan, lacto-vegetarian, and ovo-vegetarian) may provide on average between 262 mg/d and 343 mg/d of choline (at 2200 kcal) [4]. This amount is 15–50% lower than the adequate intake recommendations for pregnant and lactating women [4]. Choline is present in the diet in water-soluble forms (phosphocholine, glycerophosphocholine, and free choline) and lipid-soluble forms (sphingomyelin and phosphatidylcholine) [5]. These compounds are secreted into the human milk where they represent the majority of choline sources for the infant [6,7]. Phosphatidylcholine excreted in the bile constitutes a re-usable source of choline for the body. Several supplemental forms of choline exist such as choline chloride, choline bitartrate, and phosphatidylcholine. Multiple choline compounds are bioavailable and can support normal liver function, including choline salts and lecithin (a mixture of lipids, containing chiefly phosphatidylcholine).

## 3. Choline Physiology and Relation to Liver Function

Starting from the 1930s, studies have established that a choline-deficient diet causes fatty liver in animals and that adequate choline in the diet (e.g., lecithin purified from egg yolk, soya, or fresh beef liver, choline chloride, or choline bitartrate) can correct or prevent this phenotype [8,9,10,11]. The evidence on the role of choline in normal liver function is consistent across various species such as rats [12,13,14,15,16], mice [12], hamsters [17], pigs [18,19], dogs [20], monkeys [21,22,23], and humans [24,25]. The effect of a choline-deficient diet on fatty liver is specific and fatty liver caused by choline deficiency differs from that associated with deficiency of folate or methionine [26]. Minor between-study variations in the degree of fatty liver after eating a choline-deficient diet may be due to the compositions of the experimental diets.

Three main mechanisms can explain the role of choline in normal liver function (Figure 1). First, choline phospholipids (i.e., phosphatidylcholine and sphingomyelin) participate in lipoprotein-mediated transport of triglycerides in the body [27]. Phosphatidylcholine is necessary for the assembly of the very-low-density lipoprotein (VLDL) and thereby it participates in secretion of triglycerides from the hepatocytes (i.e., experiments in rats) [28,29,30]. Defective synthesis of hepatic VLDL particles and their secretion into the circulation could be the main mechanism involved in causing fatty liver under dietary choline deficiency conditions [26]. In line with this, a choline-deficient diet lowered phospholipid levels and the ratio of phosphatidylcholine to phosphatidylethanolamine in the liver [31]. In contrast, adequate dietary choline can maintain normal biliary phosphatidylcholine and cholesterol [32,33]. The second mechanism is related to the role of choline as a methyl donor after irreversible oxidation to betaine. Betaine can provide S-adenosylmethionie (SAM), the cofactor for the majority of cellular methyltransferases. Three SAM molecules are needed by phosphatidylethanolamine *N*-methylmethyltransferase (PEMT) to convert phosphatidylethanolamine to phosphatidylcholine in the liver [34]. Finally, choline is a source of phospholipid biosynthesis. Phospholipids are a critical component of the hepatic cell membrane that is needed for normal liver function [35].

The susceptibility to fatty liver under insufficient choline intake can be influenced by several factors (Table 1). The severity of fatty liver depends on the duration of eating the deficient diet [12,22] and the amount of fat in the diet [11,23]. Moreover, low dietary intakes of methionine or folate may deplete liver choline and may potentiate the effect of choline deficiency on fatty liver [36,37,38].

Human studies have shown that young women are less susceptible to signs of insufficient choline intake compared to postmenopausal women or men [39], which could be explained by induction of the PEMT gene by estrogen [40]. These results are in line with studies on rats showing that males are more susceptible to negative effects of a choline-deficient diet than females [41]. Also, younger animals are more sensitive to a choline-deficient diet than older ones [14,22,41], suggesting higher requirements for dietary choline and/or less endogenous synthesis in the liver of young animals compared to older ones. This suggestion is supported by studies in rats showing that liver PEMT activity at 3–4 days after birth was approximately 50% of adult liver PEMT activity and it reached adult’s activity 10 days after birth [42].

Pregnancy and lactation are associated with higher susceptibility of the mother to choline deficiency. Choline depletion in the maternal liver occurs despite 10–24% upregulation of maternal liver PEMT activity [42,43], thus highlighting the importance of adequate choline intake through the diet. 

## 4. Source of Fetal and Neonatal Choline

Only a minor amount of choline is locally produced in the placenta [44]. In vitro studies using dually perfused placenta models from Guinea pigs [45] or humans [46], and those using human placenta fragments [47], have shown that ^3^H-choline is rapidly taken up into the placenta and retained on the fetal side. The placenta tissues contain 3- to 4-fold more choline (as nmol/g tissue) than the liver of non-pregnant rats [42]. In line with this, human studies have shown that phosphatidylcholine synthesized by the PEMT pathway is enriched in the placenta and cord blood compared to maternal plasma [48].

In a controlled dietary intake study in pregnant rats, Garner et al. showed that ^14^C-labelled choline added to the maternal diet was transferred to the fetus and caused high levels of ^14^C-phosphatidylcholine and ^14^C-phosphocholine (measured as Bq/mg tissue) especially in the liver followed by the brain of the fetus [49]. A minor amount of the labelled free choline was detected in the fetus, suggesting that free choline from the maternal diet crossed the placenta and was mostly converted to phosphatidylcholine and phosphocholine in the fetus [49].

The choline levels in the neonatal rat’s liver increased between gestational days 20–22 and birth and were approximately 50% higher at birth compared to choline levels in the liver of adult rats [42], while liver PEMT activity at birth was roughly one third of that in the liver of adult rats [42]. Neonatal PEMT activity in the liver continued to increase after birth and reached its adult level at the age of 10–12 days [42]. Therefore, de novo choline synthesis via PEMT in the fetus at 20–22 days is low and does not explain high choline levels in the liver of the newborn rat compared to that in the adult rat [42]. The alternative explanation for the high choline content in the liver of the newborns would be that this extra choline must have been obtained from the mother prenatally. Similar data were reported in human studies on preterm infants (<28 gestational weeks) aged 5 days where deuterated methyl-D9-choline was incorporated into phosphatidylcholine via the cytidine diphosphocholine (CDP-choline) pathway [50], suggesting that PEMT activity in preterm infants, and possibly in the human fetal liver, is very low [50].

The presence of a fetal-to-maternal concentration gradient of choline is supported by studies showing markedly higher choline concentrations in cord plasma [51,52,53,54], amniotic fluid, and human milk [55,56,57] compared to those in the mother’s circulation.

## 5. Choline Homeostasis during Pregnancy and Lactation

Pregnancy and lactation are associated with marked changes in choline homeostasis and metabolic adaptation to choline drainage from the mother to the fetus/neonate. In rats fed a standard diet during pregnancy and lactation, maternal liver choline was depleted compared to that of non-pregnant rats fed the same diet [43,58]. Liver choline depletion during pregnancy and lactation was even more prominent when rats were fed a choline-deficient diet [58].

Liver choline is likely to be depleted in the mother in the second half of pregnancy [43,58]. In contrast, the maternal plasma concentrations of choline are high in the second half of pregnancy compared to the first half [59]. The demands for choline could be higher at late pregnancy due to active transport to the fetus, expansion of the fetus, and biosynthesis of acetylcholine in the placenta and methyl groups in the liver. In addition, pregnant women excrete larger amounts of choline in urine during the second half of gestation compared to non-pregnant women (i.e., 9 mg versus 2 mg per g of creatinine) [60]. Urinary choline excretion normalizes approximately 1 week postpartum [61]. This high urinary choline excretion in pregnant women was not influenced by increasing dietary choline intake (400 mg/d or 900 mg/d) [60]. Therefore, a physiological loss of choline (and betaine) through the kidney during pregnancy could be simply due to having more choline levels in the plasma and could partially contribute to liver choline depletion in the mother.

Depletion of maternal choline continues during lactation, especially if the maternal diet does not provide sufficient choline. The choline content in milk of lactating rats reflected the content of choline in the diet of the mother (e.g., low milk choline under choline-deficient diet, and high milk choline under choline supplementation versus the controls) [55]. A placebo-controlled trial showed that total choline concentrations were higher in maternal plasma and breastmilk from lactating women who were randomized to receive 750 mg/d of supplemental choline from 18 wk of pregnancy to 3 months postpartum compared to women in the placebo group [6].

It is possible that with the progression of pregnancy, choline could be mobilized from recent dietary or supplemental choline intake and from the maternal liver (choline derivatives stored in hepatocytes and phosphatidylcholine produced via PEMT) to the plasma of the mother. Maternal circulating choline that steadily increases in the second half of the pregnancy [59] may be considered as a ready-to-use reservoir that can be transferred to the fetus and to human milk, but partly filtered through the kidney and lost in urine of the mother. In contrast to folate where the critical time window of sufficient intake extends from preconception until the end of the third trimester, choline requirements appear to be high during the whole periods of pregnancy and lactation.

## 6. Dietary Choline Deficiency and Fatty Liver in the Mother and the Offspring

Maternal choline intake should be sufficient to satisfy the needs of the mother and the fetus or infant. Compared to a standard diet, a choline-deficient diet fed to pregnant pigs caused typical fat accumulation in liver tissues of the baby pigs ([62,63], overlapping experiments). Moreover, a choline-deficient diet in pregnant rats causes fatty liver in both the mother and the offspring [62,64,65]. A choline-deficient diet was fed to female rats starting either from 30 days before pregnancy or from the beginning of the pregnancy for the duration of pregnancy (21 days) [64]. The pregnant rats were compared to non-pregnant rats receiving the same deficient diet or a diet containing 2% choline chloride (a choline-sufficient diet) for the same duration [64]. The study reported more severe fatty liver in the mother and in the offspring when exposure to the choline-deficient diet lasted for a longer period (starting pre-pregnancy versus during pregnancy) [64]. There was a clear agreement in the degree of lipid accumulation in the liver between the choline-deficient mothers and their respective offspring [64]. Moreover, the fatty liver was more severe in pregnant rats than in non-pregnant rats fed the same choline-deficient diet [64], confirming the results of an earlier study in rats showing that pregnancy depletes liver choline and increases the susceptibility of the mother to choline-deficiency-related fatty liver [58]. Results on fatty liver in the newborn animals as a result of maternal dietary choline deficiency are in agreement with those on the effect of insufficient dietary choline intake on the liver in human adults and non-pregnant animals (discussed in Section 3).

As mentioned earlier, the young age of experimental animals makes them more susceptible to fatty liver under choline deficiency [22,37,41]. Baby pigs (aged 1–4 days) that were fed a choline-deficient diet or a choline- and folate-deficient diet for 8 weeks had a lower body weight in addition to severe fat infiltration in liver biopsies compared to pigs fed a control diet [37]. Similarly, feeding a choline-deficient diet to young rats for 6–10 days caused fatty liver, while adding choline chloride to the diet for an additional 4 days resulted in partial recovery of the fatty liver [41]. Based on these experiments, it can be argued that insufficient choline intake during human pregnancy and lactation could impact liver function in the mother, the fetus, and the infant.

A high agreement exists between human and animal studies with regard to the direction and specificity of the effect of a choline-deficient diet on development of fatty liver and the need for an external dietary source of choline to maintain normal liver function. Moreover, the mechanisms explaining fat accumulation in the liver under choline deficiency are similar across species. This justifies extrapolation of results of animal studies to humans.

## 7. Modifiers of Choline Requirements during Pregnancy and Lactation

Choline requirements in pregnancy and lactation are subject to effect modifications by the mother’s PEMT genotype [66], estrogen levels, vitamin B12 status, folate status, genotypes of folate metabolizing enzymes [66], and docosahexaenoic acid (DHA) intake [67,68,69]. These factors imply possible higher choline requirements among subgroups of women. Higher choline intake in pregnant women with low folate intake may be associated with a lower risk of neural tube defects (versus women with low choline intake) as suggested in a recent meta-analysis of case–control studies [70]. The role of choline (via betaine) as a methyl donor suggests that insufficient maternal choline intake can exacerbate the effect of nutrient deficiencies (folate or vitamin B12) or polymorphisms (e.g., methylenetetrahydrofolate reductase, MTHFR) on hyperhomocysteinemia and the risk of adverse pregnancy outcomes.

Choline endogenous synthesis via PEMT enhances incorporation of DHA into functional phospholipids such as membrane phospholipids. Thus, the PEMT genotype and choline intake could modify the association between DHA intake and fetal brain development. The interactions between the PEMT genotype and intakes of choline and DHA suggest that insufficient choline intake could counterbalance some of the advantageous effects of high maternal DHA intake on normal fetal brain development in women carriers of polymorphisms in the PEMT gene. Supplementation of 550 mg/d of choline (vs. 25 mg/d) on top of habitual choline intake across the second and third trimesters of pregnancy in women receiving 200 mg/d of DHA increased total plasma phosphatidylcholine by 16% in the intervention (vs. control) group and increased DHA status [67]. Increasing maternal choline intake beyond the current recommendations caused a 75% relative increase in maternal DHA status in the intervention arm compared to the control arm [67]. Therefore, maternal total choline intake of >550 mg/d may support the effect of DHA on fetal outcomes (e.g., normal brain and eye development of the fetus).

## 8. Implications for Future Research

Future studies need to establish whether the requirements for choline might be higher under a high intake of fat. In addition, choline requirements might differ in pregnant women with risk factors such as obesity and gestational diabetes.

Human milk contains multiple water- and lipid-soluble choline derivatives. Measuring the concentrations of these compounds is challenging, which causes large variability in results of studies from different populations. Moreover, studies are needed on a possible influence of maternal choline intake from different foods or supplements (choline salts versus phosphatidylcholine) on milk choline derivatives.

Acute fatty liver (reviewed in [71]) and non-alcoholic fatty liver disease (NAFLD) are common during pregnancy [72] and could be associated with an overweight status and fatty liver in the infants [73]. Marked changes in plasma lipid homeostasis occur during human pregnancy [74], especially in women with a high risk for gestational diabetes [75]. The reasons for these changes have not been adequately investigated, especially in relation to dietary choline intake. Moreover, the phenotype of infantile fatty liver is not commonly reported, possibly because it is not regularly investigated. The impact of insufficient maternal choline intake on liver function, obesity, or disorders in fat metabolism of the child later in life is not known.

No optimal single blood marker exists to accurately reflect variations in choline intake or status. Mendelian Randomization studies have shown that plasma concentrations of several choline-related metabolites were associated with maternal health outcomes such as gestational diabetes, body mass index during pregnancy, percent of body fat, and insulin resistance (reviewed in [76]). Surrogate markers of liver steatosis at early gestation have been shown to predict a higher risk of gestational diabetes at a later gestational age [77]. This association was largely attributed to lipid metabolites such as lysophosphatidylcholine and ceramides [77]. A possible association between choline intake at early pregnancy and the risk of gestational diabetes deserves more investigation.

Besides fatty liver, choline deficiency in animals before and during pregnancy can cause serious outcomes such as a low reproductive rate, fetal growth retardation, premature death, heart dysfunction, renal degeneration, and hemorrhage of the kidney of the fetus [41,78]. Future studies need to investigate whether insufficient choline intake could explain similar conditions in humans.

## 9. Implications for Policy and Practice

The American Medical Association (AMA, 2017) published a resolution that supports adding evidence-based amounts of choline to all prenatal vitamin supplements. Although no specific amount of choline in prenatal supplements was recommended, the AMA report implicates that adding 50 mg/d of choline is not sufficient. The Nordic Council of Ministers (represents Scandinavian countries) included choline for the first time in their newly released Nordic Nutrition Recommendations 2023 [79]. In 2023, the EFSA Panel on Nutrition, Novel Foods and Food allergens positively evaluated the first health claim proposal on maternal choline intake and health of the fetus and infants: “maternal choline intake during pregnancy and lactation contributes to normal function of the liver of the fetus and exclusively breastfed infants” [80].

Choline supplementation is considered to be safe. The Tolerable Upper Intake Level for choline is 3.5 g/d, which should not be exceeded on a daily basis [1,3]. This upper intake limit is 6- to 7-fold higher than the reference values and hardly reachable through consuming foods or common food supplements. Large doses of choline (7.5 g/d or higher) have been reported to cause hypotension, gastrointestinal problems, and fishy body odor. The current reference intakes for choline for pregnant and lactating women (450 and 550 mg/d according to NAM) [1,3] are likely to be sufficient for normal fetal development in uncomplicated pregnancies. However, the majority of young women consume approximately 300 mg/d of choline, which is 150–250 mg/d lower than the reference intake values [81,82]. Therefore, public health policies are needed to ensure sufficient choline intake through adding choline to maternal multivitamin supplements.

## 10. Conclusions

Choline requirements are higher during pregnancy and lactation than in non-pregnant women. During these critical life stages, the liver of the mother becomes depleted of choline and at the same time choline accumulates in the placenta and the liver of the fetus or is excreted in human milk. There is a mother-to-child gradient that may predispose the mother to choline deficiency and may cause fatty liver in the mother and the child. A sufficient choline intake during pregnancy and lactation (450 and 550 mg/d according to NAM) contributes to normal liver function of the fetus and breastfed infants. More studies are needed on whether these intake levels may influence additional health outcomes in the fetus and infant. The gap between the present intake recommendations and the practice is huge. Current knowledge suggests that adding up to 250 mg of choline to maternal multivitamin supplements should be a safe and an effective way to help in bringing the average choline intake of pregnant and nursing mothers closer to the recommended amount.

## Figures and Tables

**Figure 1 nutrients-16-00260-f001:**
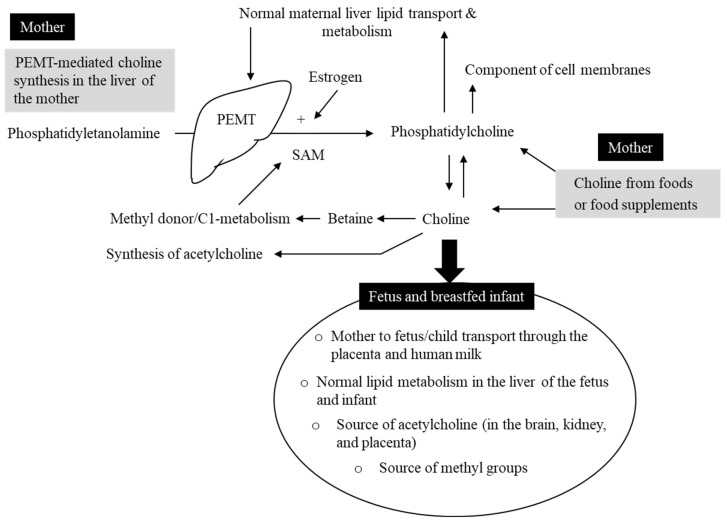
Choline hemostasis during pregnancy and lactation. Choline is provided by foods or food supplements in the form of water-soluble or fat-soluble choline derivatives. A small amount of phosphatidylcholine is produced in the liver from phosphatidylethanolamine via phosphatidylethanolamine *N*-methylmethyltransferase (PEMT). PEMT enzyme requires methyl groups provided by S-adenosylmethionine (SAM). High estrogen during pregnancy and lactation leads to higher PEMT gene expression and stimulation of phosphatidylcholine synthesis in the liver of the mother. Phosphatidylcholine is an essential component of cell membrane and lipoproteins that remove triglycerides from the hepatocytes and transport them in the blood. Choline is also a methyl donor and a substrate for producing the neurotransmitter acetylcholine. Choline is actively transported to the fetus through the placenta and to the breast-fed infant through human milk. Insufficient choline in the diet of the mother can lead to depletion of choline stores in the liver of the mother and accumulation of triglycerides in the liver of the mother and the offspring.

**Table 1 nutrients-16-00260-t001:** Risk factors that can increase the susceptibility to negative health outcomes of low dietary choline intake.

Risk Factor	Explanation
Men (vs. women)	Men are more susceptible to fatty liver under low choline intake because PEMT gene expression is induced by estrogen in women (higher PEMT-mediated choline production in women than in men)
Postmenopausal women (vs. pre-menopausal)	Young women have higher endogenous production of choline due to the effect of estrogen on PEMT
Newborns and infants	Liver PEMT activity is low at birth and the demands for choline are higher than in adults
Pregnancy	High demands compared to non-pregnant women and active transfer of choline to the fetus can deplete choline from the liver of the mother and predispose her for choline deficiency (e.g., fatty liver)
Lactation	High excretion of choline derivatives into breastmilk may deplete choline from mother’s liver and predispose her for choline deficiency (e.g., fatty liver)
Low vitamin B12 or folate intake or MTHFR 677 TT genotype	Adequate folate and vitamin B12 support choline endogenous production by providing S-adenosylmethionine needed for PEMT enzyme
High-fat diet	Triglycerides accumulate in the liver if choline intake is not proportional to fat content in the diet. Adult Wister rats fed a choline-deficient and fat-rich (40%) diet developed fatty infiltration of the liver within 21 days [11]. Supplementing the high-fat diet with 50–70 mg of choline daily reduced fat content in the liver of the animals that were previously fed a choline-deficient diet. Choline also prevented further accumulation of fat in the liver under continuous high-fat diet [11]. These results suggest that dietary fat intake could determine choline requirements
High sugar intake, toxins such as alcohol	These factors decrease the ability of the liver to metabolize fats and enhance fatty liver
Polymorphisms in PEMT gene (e.g., PEMT rs7946)	Carriers of some genotypes could have lower PEMT activity, implying higher requirements for dietary sources of choline
A plant-based diet	Adherence to a lacto-vegetarian or a vegan diet provides up to 50% lower dietary intake of choline compared to the adequate intake levels for pregnant and lactating women [4]. In addition, low choline and B12 intake in the same time can challenge one-carbon metabolism (e.g., hyperhomocysteinemia). Women adhering to a vegetarian or a vegan diet are at risk of insufficient choline intake and a target group for choline supplementation
The duration of a low-choline diet	A choline-deficient diet (e.g., contains 50 mg/d of choline) can cause liver damage (elevated liver enzymes) within 3 weeks in male subjects

## Data Availability

Data are contained within the articles.

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
