# Peer review of "A Narrative Review on Maternal Choline Intake and Liver Function of the Fetus and the Infant; Implications for Research, Policy, and Practice"

_nutrients, 2024, doi:10.3390/nu16020260_

Round 1

Reviewer 1 Report

Comments and Suggestions for Authors

The article is well-written, and the authors discuss a very relevant topic. Choline intake & choline deficiency during pregnancy and lactation is an emerging issue that has been neglected so far.

Suggestions:

- Please clearly declare the type of article ("narrative review") in the title.

- Regarding the poor intake of choline in pregnant women following a vegan diet, an additional paragraph should be added discussing the relevance of the findings for (pregant/breastfeeding) vegans.

Author Response

Please see the point-by-point reply in the attachment

Reviewer 2 Report

Comments and Suggestions for Authors

The issues are relevant to the development trends of academic research and the needs of clinical care. The overall literature review can also significantly highlight the importance of the topic that should be highly valued.  It is necessary to clarify some minor issues.

1. There are too many keywords. In particular, it is recommended to delete keywords that do not appear in the abstract or appear too low in this article.

2. Line 67-74. It is recommended to indicate how many documents were found in total?

3. Although it is very important to supplement with adequate amounts of choline, it is recommended to briefly explain: What are the side effects if excessive choline is taken.

4. It is recommended to add a "Conclusion" paragraph.

Author Response

(The authors gave the same response as above.)

Reviewer 3 Report

Comments and Suggestions for Authors

Maternal Choline Intake during Pregnancy and Lactation and 2 Liver Function of the Fetus and the Infant; Implications for 3 Research, Policy and Practice

Summary: The authors present a review of the effects of choline deficiency during pregnancy on the development of fatty liver disease. Unfortunately, it does not present a compelling rationale for doing the review or any compelling conclusions.

Specific comments:

1.      Abstract: There seems to be a logical inconsistency here. The severity of disease was more severe with a choline deficit prior to pregnancy, so while maternal choline intake contributes to the normal development of liver, shouldn’t there also be concern with choline intake prior to becoming pregnant? Such that a woman planning to become pregnant should have dietary guidelines as well? If the focus is strictly on choline during pregnancy, then I suggest removing the comment about severity being worse if the diet is started pre-pregnancy and putting that elsewhere in the manuscript.

2.      Overall, what is the purpose of this review? Is it to inform public policy? Is it to suggest new avenues of research? It would be beneficial to have an overarching purpose to tie the paper together. As written, it seems to be a list of interesting, but not necessarily related (other than pertaining to choline) factoids.

3.      Page 2, line 48-50. It would be helpful if the dietary recommendations for non-pregnant women were presented in this paragraph.

4.      Paragraph 2, lines 51-63. This seems rather disorganized. It starts with food sources of choline, but then transfers into choline absorption and transport before veering back to choline supplements.

5.      Paragraph 3 is confusing. If research since 1990 does not answer the question, why did the literature search for maternal choline effects stop at 1990? Wouldn’t there be some literature post-1990 that also addresses the effects of maternal choline deficits? What was the rationale for 1990?

6.      Page 2, lines 84-94. The authors say that there are three mechanisms for the role of choline in normal liver function. They devote 1 sentence to the first and the remainder of the paragraph to the second. The third gets a paragraph to itself. This is very uneven treatment. How, for example, would decreased biosynthesis of phospholipids impact normal liver function?

7.      Organization throughout is an issue. Paragraphs tend to be about 3 sentences of disjointed factoids. There is no topic sentence or meat in the paragraph. An example paragraph is lines 122-16. This starts with choline supply from the mother, then transitions to placental production, and then transitions back to placental uptake. Pick one, discuss that thoroughly and then move on to the next. Perhaps start with the lack of placental production, which would require that choline come from somewhere else. Then talk about plasma choline levels and focus on placental transfer from plasma to fetus (placental uptake).

8.      There are minor grammatical errors throughout. E.g., line 130 “14C-labled choline …was transferred”, not “were transferred”. Some of the material in this paragraph would be nicely paired with the preceding paragraph and strengthen the paper as well as improve the organization.

9.      Line 145: How do the authors explain prenatal exposure contributing to maximal liver choline at 3-4 days postnatal? I would think that milk would be the source of the choline 3-4 days postnatal with prenatal exposure effects waning.

10.   Why would pregnant women excrete more choline during the second half of gestation? This doesn’t suggest to me that demands for choline are increased, but rather that choline is being over-synthesized and excess choline is excreted. What other experiments suggest high demand rather than over production?

Comments on the Quality of English Language

Minor editing for grammar is required. Some verb tenses are incorrect.

Author Response

(The authors gave the same response as above.)
